# Clinical efficacy and safety of automatic remifentanil administration based on Analgesia Nociception Index monitoring during burn surgery under propofol anesthesia: A randomized controlled clinical trial

Maxence Hureau[1,2,3], Emeline Caillau[4], Julien Labreuche[4], Mathilde Herbet[2], Benoît Tavernier[1,5], Julien De Jonckheere[2,5]*, Mathieu Jeanne[1,2,3]

1 CHU Lille, Anesthesia and Critical Care, Lille, France, 2 CHU Lille, CIC-IT 1403, Lille, France, 3 Univ. Lille, ULR 7365 - GRITA, Lille, France, 4 CHU Lille, Statistique, Evaluation, Economique, Data-management Lille, Lille, France, 5 Univ. Lille, ULR 2694 – METRICS, Lille, France

* julien.dejonckheere@chu-lille.fr

## Abstract

### Background

Monitoring the nociception/antinociception balance for analgesic guidance during general anesthesia may improve the quality of anesthesia. The University Hospital of Lille (France) has developed an expert software system for automatic remifentanil administration based on the continuous monitoring of the Analgesia Nociception Index (MDoloris, France). We assessed the clinical efficacy and safety of the ANI-REMI-LOOP "expert-system software" during burn surgery in a monocentric randomized controlled trial.

### Methods

The trial was approved by the French Ethics Committee, and all patients gave written informed consent. From 2018 to 2022, 52 adults were randomized into two groups: manual remifentanil infusion (standard practice) or automatic remifentanil infusion (expert-system software) during BIS-guided propofol anesthesia at the burn center of the hospital. In the standard practice group, remifentanil administration was based on Minto's model and guided by the analgesia nociception index. In both groups, propofol was administered based on Schnider's model and guided by the BiSpectral Index (Covidien). The primary endpoint was the cumulative remifentanil dose administered during anesthesia and secondary endpoints were related to the clinical safety of automatic remifentanil administration with the incidence and duration of hypotension, bradycardia, hypertension or tachycardia related to nociception. After anesthesia, the

**Data availability statement:** All relevant data are within the paper and its Supporting Information files.

**Funding:** This research was funded by a grant of € 17500 from the APICIL Foundation (Lyon, France). The funders had no role in study design, data collection and analysis, decision to publish, or preparation of the manuscript.

**Competing interests:** "J.D.J. and M.J. are scientific advisers for and own shares of MDoloris Medical Systems, Loos, France. M.Hu., E.C., J.L., M.He,. and B.T. declare.

endpoints were pain and analgesic requirements during 2 hours. A p value < 0.05 was considered statistically significant. Data are presented as median [1st to 3rd quartile].

## Results

The cumulative remifentanil dose was significantly lower in the automatic group 0.125 µg.kg$^{-1}$.min$^{-1}$ [0.106 to 0.149] vs. 0.152 µg.kg$^{-1}$.min$^{-1}$ [0.137 to 0.237], $p = 0.004$), and the cumulative proportion of time with hemodynamic impairment or reactivity was significantly lower in the expert-system automatic group 4.2% [2.5 to 5.7] vs. 19.4% [6.9 to 59.9], $p = 0.010$). There were no safety issues, and pain and analgesic requirements were similar in both groups after surgery.

## Conclusions

Automatic remifentanil administration demonstrated good clinical performances during propofol anesthesia for burn surgery. It is likely that these results can be extrapolated to any surgical setting under general anesthesia, but this needs to be tested with further randomized clinical trials.

## Introduction

Monitoring unconsciousness and myorelaxation is standard practice during anesthesia, but monitoring antinociception is still a field of active research. Several monitors of the autonomic nervous system (ANS) have shown good performances in assessing the nociception/antinociception (NAN) balance, a term commonly used to describe the dynamic interaction between nociceptive stress induced by the surgical procedure and the antinociceptive action of analgesic drugs administered during general anesthesia. Modern anesthesia has turned to monitoring ANS reactions to monitor indirectly nociception, which leads to sympathetic activation when left unchecked, while adequate antinociception enhances parasympathetic (paraS) activity. Therefore, "nociception monitoring" relies nowadays on ANS monitors displaying measurements of sympathetic or paraS tones which act as surrogates for NAN balance monitoring during general anesthesia [1]. The University Hospital of Lille (France) has developed the Analgesia Nociception Index (ANI; currently commercialized by MDoloris Medical Systems, Lille, France), which measures real-time heart rate variability (HRV) related to paraS activity [2–5]. Independent clinical trials have demonstrated that ANI values are a valid surrogate for NAN balance monitoring [6–10], safe for opioid guidance during general anesthesia [11–19], unaffected by surgical settings and by relative hypovolemia [20].

Several authors have described automatic opioid delivery during general anesthesia, but the lack of a clear physiological nociceptive signal has led to indirect assessments of nociception, often based on pharmacodynamic interaction models, hemodynamic reactions, or electroencephalographic signals related to cortical arousal

[21–23]. In most studies, automated delivery of propofol and remifentanil was shown to be effective and safe, even if the delivery control strategies were rather simplistic, based on proportional, integral, derivative (PID) controllers, or controlling two pharmacological compounds (typically hypnotics and opioids) with a single input variable derived from a simplified EEG signal (e.g., BiSpectral® index) which is neither sensitive nor specific of nociception. In order to improve the performance of remifentanil automatic administration, we have recently designed an expert system for the automatic administration of remifentanil during general anesthesia, whose decision rules for modifying remifentanil flow rate are based on two objectives: i) maintaining stable hemodynamics by avoiding hypertension or tachycardia related to nociception as well as avoiding hypotension and bradycardia and ii) maintaining a relative predominance of paraS activity as assessed by ANI by dynamically increasing or decreasing the remifentanil infusion rate [24,25,26].

At the Burn Center of the University Hospital of Lille, ANI guidance of multimodal antinociception during general anesthesia is standard practice for excision and skin graft surgery of burn patients. Our hypothesis was that remifentanil could be automatically administered during burn excision–skin graft surgery with an expert-system software. We thus designed a randomized clinical trial to assess the efficacy and safety of a custom-made expert system, the "ANI-REMI-LOOP software", for automatic remifentanil administration during propofol anesthesia vs. remifentanil manual infusion (standard practice).

The objectives of the study were to evaluate the potential benefit of automatic remifentanil administration on remifentanil or propofol consumption, hemodynamic stability, ANI stability and postoperative pain.

## Materials and methods

Several randomized clinical trials have been carried out in various surgical settings to evaluate the benefit of guiding the administration of remifentanil with nociception/antinociception balance monitors [14–19]. The designs and objectives of these studies are relatively standardized, we therefore designed the methodology of our clinical trial on the existing literature.

The French Ethics Committee (Comité de Protection des Personnes – Ile de France V) approved the trial, number 2017-A00858-45, which was registered with ClinicalTrials.gov (NCT03556696) [27]. All patients were included between June 27, 2018, and June 14, 2022 and gave written informed consent. Patients were randomized into either expert-system automatic remi infusion or manual remi infusion (standard practice) groups using a prerecorded allocation list of permutated blocks of four (Ennov Clinical®, Paris, France). The patients were blind to their allocation.

### ANI monitor and ANI-REMI-LOOP system

ANI monitors of MDoloris technology perform real-time HRV analysis on time windows of 64 s, band-passing the RR-series in the high-frequency field for paraS assessment. The normalized paraS activity is expressed as ANI on a 0–100 scale, indicative of paraS relative predominance when ANI values are superior to 50, and paraS withdrawal when ANI values are below 50 [3,9]. Technically, each 64 s window yields one ANI measure, which is stored in a stack of ANI values if the signal quality is good. Two averaged values are displayed continuously: one "short average" called "instant ANI (ANIi) measured over 120 s and one "long average" called "average ANI (ANIm) measured over 240 s.

The ANI-REMI-LOOP, a computer running the custom-made expert-system is connected to an ANI monitor (in this study, the PhysioDoloris, MDoloris Medical Systems) and a standard infusion pump for remifentanil infusion (Alaris GH Medical, Hampshire, United Kingdom) via RS232 serial ports. The computer is also connected to the GE Datex-Ohmedia S/5 monitor (Chicago, Illinois, USA) which is used for anesthetic monitoring, so that the expert-system processes heart rate (HR), systolic blood pressure (SBP), ANIm and ANIi. All four parameters are displayed continuously for supervision (see supplemental digital content 1; details have been published elsewhere [24,25,26]). Fig 1 presents an overview of the implemented hardware and the ANI-REMI-LOOP with its connections. Briefly, in "manual mode," the anesthetist determines the flow rate of the electrical syringe by interacting manually with the software, which can also deliver a remifentanil bolus of 1 µg.kg$^{-1}$ for example at the start of anesthesia.

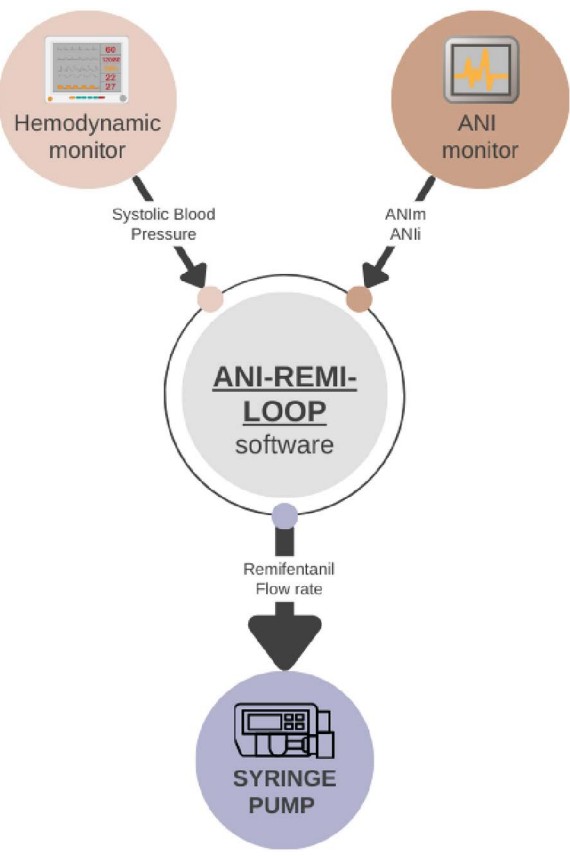

**Fig 1. Schematic presentation of the ANI-REMI-LOOP software with inputs and outputs.**

When switched into "automatic mode," the expert system follows simples rules based on the input variables ANIi, ANIm and SBP, and adapts dynamically the remifentanil infusion rate. Expert rules are aimed at i) maintaining a predominance of paraS activity: a range of [50–75] for ANIm is targeted by increasing the remifentanil flow rate when ANIm is below 50 or decreasing it when ANIm is over 75. When a rapid decrease of ANIi occurs, which indicates paraS withdrawal and can be interpreted as an offset of the NAN balance by a rapid increase in nociception, the remifentanil flow rate is increased even if ANIm is still in the [50–75] target range; ii) avoiding hypotension: if SBP is lower than 80 mmHg, then remifentanil flow rate is stopped. If SBP is decreasing and within the [80–100] mmHg range, then the remifentanil flow rate is decreased proportionally to the SBP decrease.

**Expert-system description**

**Definitions of infusion flow rate and bolus.**

• **Infusion flow Changes:** Increase or decrease the syringe pump infusion rate.

• **Bolus:** short and fast increase of infusion rate during 10 s (infusion flow returns to its previous value after these 10 s).

Each infusion flow change or Bolus is immediately followed by a refractory period of 10 s (see below) during which no further change in infusion flow or Bolus can occur.

**Definitions of regulation variables and constants.** Variables:

- **$ANI_i$**: instantaneous ANI

- **$ANI_m$**: mean ANI

- **$S_i$**: slope of $ANI_i$ over 30 s (rad)

- **$S_m$**: slope of $ANI_m$ over 30 s (rad)

- **InF**: Syringe pump infusion Flow (in µg/Kg/min)

- **Bolus**: Infusion Flow bolus increment (µg.kg$^{-1}$.min$^{-1}$) = InF * 2 + 0.0375

- **SBP**: Systolic blood pressure (mmHg)

- **$SBP_p$**: Previous SBP value (mmHg)

- **HR**: Heart Rate (bpm)

Constants:

- **$ANI_{max}$**: High threshold for the target range of $ANI_m$ = 75

- **$ANI_{min}$**: Low threshold for the target range of $ANI_m$ = 50

- **$ANI_{mean}$**: computed as $ANI_{mean} = ANI_{min} + (ANI_{max} - ANI_{min})/2 = 62.5$

- **$Sm_T$**: Low threshold for $S_m$ = - 7

- **$Si_T$**: Low threshold for $S_i$ = -25

- **InI**: Infusion Rate increment (µg.kg$^{-1}$.min$^{-1}$) = 0.01

- **$InF_{min}$**: Minimum infusion rate (µg.kg$^{-1}$.min$^{-1}$) = 0.1

- **$Inf_{max}$**: Maximum infusion rate (µg.kg$^{-1}$.min$^{-1}$) = 1.2

- **RefP**: Refractory period = 10 s

- **$SBP_{min}$**: Security Threshold for SBP = 80 mmHg (SBP below $SBP_{min}$ lead to a stop of infusion flow)

- **$SBP_{mean}$**: Lowest accepted SBP = 100 mmHg (SBP below $SBP_{mean}$ lead to a reduction of infusion flow)

- **$HR_{min}$**: Lowest accepted HR = 45 (HR below $HR_{min}$ lead to a stop of infusion flow)

**Definition of priorities for expert rules.**

- Priority 1 rules are designed for the patient's safety and apply first. They can lead to infusion flow reductions or full stop, and cannot be bypassed.

- Priority 2 rules are designed for the dynamic adaptation of infusion flow guided by ANI measures. They apply insofar priority 1 rules are respected.

**Priority 1 rules: Hemodynamic safety.** These rules have been implemented in order to limit the risk of severe hypotension (SBP < 70 mmHg). They only allow limiting the infusion flow in case of SBP decrease.

- When SBP is higher than $SBP_{mean}$, no limitation to infusion flow applies. When SBP is comprised between $SBP_{min}$ and $SBP_{mean}$ and SBP is lower than the previous SBP value, a proportional reduction of infusion flow is applied according to the following formula:

**If** $(SBP < SBP_{mean})$ **and** $(SBP < SBP_p)$ **then** $InF = InF - InF*(SBP_p-SBP)/(SBP_p-SBP_{min})$

- When SBP is lower than $SBP_{min}$ or HR is lower than $HR_{min}$, then infusion is stopped.

    **If** $(SBP < SBP_{min})$ **or** $(HR < HR_{min})$ **then** $InF = 0$

- Each infusion rate change is followed by a RefP refractory period.

**Priority 2 rules: dynamic adaptation of remifentanil flowrate.** Several studies have demonstrated that variations in ANI enable to anticipate hemodynamic reactivity in relation with the NAN balance. We therefore implemented ANI based decision rules in order to anticipate and possibly avoid hemodynamic reactivity by increasing the infusion flow in order to maintain $ANI_m$ slightly over $ANI_{min}$. Other rules also aimed at decreasing the infusion flow in order to maintain $ANI_m$ bellow $ANI_{max}$.

Acute parasympathetic withdrawal, expressed as a decrease of $ANI_i$ below its low threshold with a slope $(S_i)$ below its threshold

- **If** $(ANI_i < ANI_{min})$ **and** $(S_i < S_iT)$ **then** Bolus

    Each Bolus is followed by a 3*RefP s refractory period.

    Slow parasympathetic withdrawal, expressed as a decrease of $ANI_m$ below its low threshold with a slope $(S_m)$ below its threshold:

- $ANIm \leq ANImin$:

    Increase infusion flow.
    **If** $(InF < InF_{max})$ **and** $(S_m < S_mT)$ **then** $InF = InF + InI$
    Parasympathetic overshoot, expressed as an increase of ANIm over its high threshold:

- $ANIm \geq 75$:

    Decrease infusion flow.
    **If** $(S_m > 0)$ **and** $(InF - InI > InF_{min})$ **then** $InF = InF - InI$
    ANIm in its target range:

- $ANImin \leq ANIm \leq ANImax$

    Infusion flow is adapted based on slope $(S_m)$ changes:
    **If** $(ANI_m < ANI_{mean})$ **and** $(S_m < S_mT)$ **and** $(InF + InI < InF_{max})$ **then** $InF = InF + InI$
    **If** $(ANI_m > ANI_{mean})$ **and** $(S_m > S_mT)$ **and** $(InF - InI > InF_{min})$ **then** $InF = InF - InI$

**Anesthesia protocol**

ASA I–III adult patients scheduled for excision of burns and skin grafts at the Burn Center (University Hospital of Lille, France) were included. Exclusion criteria were non-sinus cardiac rhythm, pacemaker, known dysautonomia, diabetes mellitus with micro or macro angiopathic complications, body mass index below 17 or over $35 \, kg.m^{-2}$, allergy to a protocol's medication, pregnancy or breastfeeding, and patients unable to consent. Baseline $HR > 120$ bpm, baseline $SBP < 75$ mmHg or baseline $SBP > 160$ mmHg led to exclusion. During anesthesia, sustained tachycardia ($HR > 120$ bpm), impaired blood pressure ($SBP < 75$ mmHg) or prolonged hypertension ($SBP > 160$ mmHg) despite adequate appliance of the anesthetic protocol led to exclusion from the per-protocol analysis. Patients' baseline HR and blood pressure were measured a few days before surgery and used to establish HR and SBP thresholds of hemodynamic reactivity during anesthesia as a 20% increase over baseline values (secondary endpoint). The patient ideal body weigth (IBW) was calculated and used for all drug dose adaptations.

Upon arrival in the operating room, three-lead ECG, pulse oximetry, and non-invasive blood pressure monitoring (GE Datex-Ohmeda S/5 monitor) were connected to the patient. A PhysioDoloris® monitor was connected to the ECG analogue output of the multiparameter monitor. A BIS electrode was positioned on the patient's forehead and connected to a BIS monitor (BIS Vista™ monitor, Covidien, Boulogne-Billancourt, France). BIS, ANIm, and ANIi were displayed continuously in both groups.

An intravenous catheter was inserted, and Ringer Lactate 500 mL was started. Midazolam (0.08 mg.kg⁻¹, max 5 mg) was administered IV, and oxygenation was started. The propofol cerebral concentration (Ce) target was set at 5 µg.mL⁻¹ (Schnider model, Orchestra Base Primea, Fresenius-Kabi, Germany) and increased stepwise until loss of consciousness [28]. Cisatracurium 0.15 mg.kg⁻¹ was administered for tracheal intubation, tidal volume was set at 8 ml.kg⁻¹ of (IBW) and respiratory rate was set at 12 min⁻¹ (GE Aisys anesthesia machine), and then adapted for maintaining the end-tidal carbon dioxide in the [30–35] mmHg range. Following orotracheal intubation, propofol Ce target was lowered to 3 µg.ml⁻¹. The propofol Ce target was then adapted to maintain BIS in the [40–60] target range by Ce target increases of 1.0 µg.ml⁻¹ (BIS > 60) or 2.0 µg.ml⁻¹ (BIS > 80), or Ce target decreases of 0.5 µg.ml⁻¹ (BIS < 40) or 1.0 µg.ml⁻¹ (BIS < 25). After each target change, the next change was not allowed until Ce reached its new intended target. Upon the end of surgery, protocolized pre-emptive analgesia was subcutaneous infiltration of the skin sampling area with 20 mL ropivacaine (2 mg.ml⁻¹), paracetamol (1 g IV) and sufentanil (0.12 µg.kg⁻¹ IV), which were all administered at the end of surgery. Ondansetron (4 mg IV bolus) was administered systematically before the end of anesthesia. In the post anesthesia care unit (PACU), analgesia was IV morphine titration (limited to a cumulative dose of 12 mg) and IV ketamine (20 mg) as rescue (persistent VAS ≥ 4/10 despite maximal morphine titration), as is customary in the unit.

## Remifentanil administration

In the standard practice group, remifentanil was started simultaneously with propofol via target-controlled infusion (Minto model, Orchestra Base Primea, Fresenius-Kabi, Germany) [29] with a Ce target of 6 ng.mL⁻¹. Following orotracheal intubation, remifentanil Ce target was lowered to 3 ng.mL⁻¹ and was then adapted to maintain ANIm in the [50–70] range by 2.0 ng.ml⁻¹ (ANIm < 35) or 1.0 ng.ml⁻¹ (ANIm < 50) increases, or 0.5 ng.ml⁻¹ (ANIm > 70) or 1.0 ng.ml⁻¹ (ANIm > 90) decreases. After each change, the next change was not allowed until Ce reached its new target.

In case of hypotension or bradycardia, remifentanil target was lowered to zero and ephedrine was administered to restore blood pressure; if insufficient, a continuous administration of diluted norepinephrine (16 µg.mL⁻¹) was started, as is standard in our unit.

In the expert-system automatic group, remifentanil was started simultaneously with propofol using the manual mode for administering a bolus of 1 µg.kg⁻¹ followed by a constant flow of 0.10 µg.kg⁻¹.min⁻¹. Following orotracheal intubation, the remifentanil flow rate was kept at 0.10 µg.kg⁻¹.min⁻¹ and manually adapted to maintain ANIm in the [50–70] target range, until initiating surgical stimulation at which point the ANI-REMI-LOOP was switched to "automatic mode." During the manual mode, remifentanil output could be manually decreased in case of hemodynamic impairment or increased in case of hemodynamic reactivity or when ANIm would decrease below 50. An investigator-engineer (J.D.J.) was present only to guarantee the patients' safety in the event of a malfunction of the ANI-REMI-LOOP software. The engineer did not act on the management of anesthesia nor influence the decisions of the anesthetists (M.J. or M.Hu.).

## Outcomes

The primary outcome was the total amount of remifentanil administered (normalized on the patient's weight and anesthesia duration). Secondary outcomes, all measured during anesthesia duration, were the total amount of propofol (normalized on weight and administration duration), the number of changes in propofol and remifentanil flow rates. In addition, a composite hemodynamic endpoint was assessed, aiming at measuring hemodynamic stability during anesthesia—the proportion of anesthesia time the patient spent in a state of either "hemodynamic reactivity," defined as a 20% increase

of HR or SBP as compared to baselines measures, or "impaired hemodynamic status," defined as hypotension (SBP < 75 mmHg) or bradycardia (HR < 40 min$^{-1}$). The said composite hemodynamic variable was titled "hemodynamic reactivity, hypotension, or bradycardia,"and the proportion of time spent in such a state during anesthesia was measured. The total amount of ephedrine, the proportion of time for ANI$_i$ and ANIm spent at under 50, in the [50–70] interval, and above 70, for BIS spent at under 40, in the [40–60] interval, and above 60 were also measured. Several time points were defined: (T1) 5 minutes before start of surgical stimulation, (T2) 1 minute, (T3) 5 minutes, and (T4) 20 minutes after initiating stimulation, (T5) end of surgery, (T6) during wound dressing, and (T7) patient unstimulated before the awakening phase. "Stimulation" was defined as the beginning of the second time iodine was applied because of the acute stimulation it induces. In the expert-system automatic group, the number of times the automatic mode was stopped and the duration of and reason for the interruption were recorded. In the PACU, pain assessed by VAS between H0 and H2, total amount of morphine and ketamine administered, and incidence of nausea/vomiting were recorded.

## Statistical analysis

Only patient's ideal body weigth (IBW) was used for dose adaptations. We determined that the data of 52 patients (26 per group) would have 80% power to detect, with a two-sided test at 5% significance level, a 20% decrease of remifentanil in the expert-system automatic group vs. the standard practice group, by considering a mean ± standard deviation of remifentanil in the standard practice group of 0.30 ± 0.075 µg.kg$^{-1}$.min$^{-1}$. Quantitative variables are expressed as mean (standard deviation) or median (interquartile range) for non-Gaussian distribution. Categorical variables are expressed as frequencies and percentages. Normality of distribution was assessed graphically and with the Shapiro–Wilk test.

Primary efficacy analysis was done on all randomized patients following their assigned intervention group and according to an intention-to-treat (ITT) analysis, and was conducted using a Mann–Whitney U test due to non-Gaussian distribution. A sensitivity efficacy analysis for primary outcome only was done in the per-protocol population after excluding patients with major deviations. The treatment effect size (with its 95% confidence interval, CI) was estimated by calculating the Cohen's standardized difference on rank-transformed data. Secondary binary outcomes were compared between the two groups using the Chi-square test, and relative risks with their 95% CIs were calculated as effect size. All secondary quantitative outcomes were compared using the Mann–Whitney U test, and standardized differences with their 95% CIs on rank-transformed data were reported as effect size. Propofol target changes were compared using a negative log-binomial regression model with log transformed anesthesia duration as the offset variable. Statistical tests were 2-sided and $p < 0.05$ was considered statistically significant. The SAS software package was used (release 9.4, SAS Institute, Cary, NC).

## Results

Between June 27, 2018, and June 14, 2022, 52 patients were randomized, 26 in each group. Two patients in the standard practice group presented with sustained ectopic beats or tachycardia and were excluded from the per-protocol analysis (Fig 2). Patients' characteristics are presented in Table 1.

The median total remifentanil amount was significantly lower in the expert-system automatic group at 0.125 µg.kg$^{-1}$.min$^{-1}$ vs. 0.152 µg.kg$^{-1}$.min$^{-1}$ in the standard practice group, with a large effect size (standardized difference = −0.88), corresponding to 17.8% dose reduction (Table 2). The significant reduction in opioid delivery most probably resulted from the specific algorithm fine-tuning remifentanil delivery to the dynamic changes of ANI.

Furthermore, the automatic group exhibited significantly reduced time spent with hemodynamic reactivity while other secondary outcomes showed no significant differences between groups (Table 3).

The median propofol administered was not significantly different between the groups with 0.115 mg.kg$^{-1}$.min$^{-1}$ in the expert-system automatic vs. 0.111 mg.kg$^{-1}$.min$^{-1}$ in the standard practice group. Similar results were observed for the number of propofol target changes, and for the time BIS spent below its target, in its target range, or above its target (Table 3

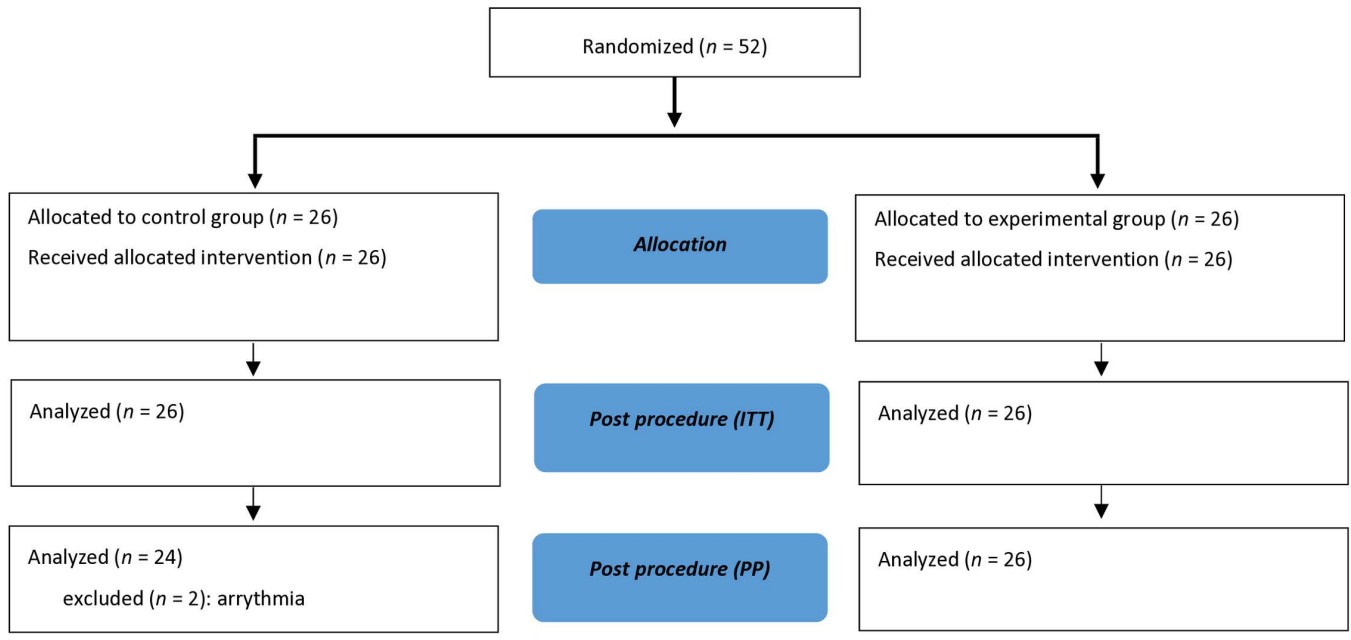

**Fig 2. Flowchart.**

and Supplemental Digital Content Table 5 in S1 File). Fig 3 presents the time course of HR, SBP, BIS, and ANIm, which were similar in both groups.

The median [1st to 3rd quartile] number of changes of remifentanil flow rate was 57 [24–100] in the expert-system automatic group and 7 [4.5–14] in the standard practice group. In 17 patients (65%), no disengagement of the automatic mode was needed, while one disengagement was needed in 5 patients (19%) to add a remifentanil bolus due to reactivity, and two disengagements were needed in 4 patients (16%). Twenty-one "Hemodynamic reactivity, hypotension, or bradycardia" events occurred in 12 cases (46.2%) in the expert-system automatic group and 27 "Hemodynamic reactivity, hypotension, or bradycardia" events occurred in 8 (30.8%) cases in the standard practice group (p=0.25). The median duration for one event was 1 [1–2] min for the expert-system automatic group and 2 [1–6] min for the standard practice group (p=0.038), and their cumulative duration was 2 [2–3] versus 10.5 [3.75–25] min respectively (p=0.005). Two hypotension and no bradycardic events were observed in the expert-system automatic group whereas 4 hypotension and 3 bradycardic events occurred in the standard practice group (Table 3).

Post-anesthesia follow-up is presented in Table 4 (and Supplemental Digital Content Table 6 in S2 File). There was no difference in pain or analgesic requirements between the two groups.

## Discussion

This monocentric RCT validated the clinical safety of an ANI-based automatic remifentanil infusion during general anesthesia and reflected a significant decrease of remifentanil administration vs. standard practice. Ephedrine use and propofol administration were similar in the two groups. Hemodynamic control was significantly better in the automatic group, and no software malfunction was reported. External interruptions of the "automatic mode" of the software were warranted by clinical reactivity in 9 patients (35%) to administer a remifentanil bolus. Pain scores and analgesic requirements at the 2-h follow-up in PACU were similar in both groups.

The significantly lower remifentanil dose in the automatic group demonstrates the automatized regulation efficacy; the small additional remifentanil boli that were automatically and regularly administered did not lead to an overall remifentanil increase, possibly due to the dynamic down-adjustments when ANIm was greater than 70.

**Table 1. Patient characteristics.**

| | Control group (n=26) | Automatic group (n=26) |
|---|---|---|
| **Demographic** | | |
| Male | 22 (84.6%) | 15 (57.7%) |
| Age (years) | 45.4±12.7 | 43.4±14.4 |
| Weight (kg) (real) | 77.1±12.2 | 73.8±14.5 |
| Height (cm) | 174.6±7.6 | 171.2±8.7 |
| Active Smoker | 10 (38.5%) | 12 (46.2%) |
| **Clinical** | | |
| Burn depth | | |
| 2nd degree | 19 (73.1%) | 18 (69.2%) |
| 3rd degree | 7 (26.9%) | 8 (30.8%) |
| Delay between burn and procedure, (days) | 18.0 (14.0–21.0) | 18.0 (16.0–27.0) |
| Excision surface (% of Total Body Surface) | 3.0 (1.5–3.5)[1] | 2.0 (2.0–3.0)[2] |
| ASA score | | |
| I | 10 (38.5%) | 15 (57.7%) |
| II | 16 (61.5%) | 10 (38.5%) |
| III | 0 (0.0) | 1 (3.8%) |
| Baseline HR | 74.5 (62.0–78.0) | 75.0 (68.0–80.0) |
| Baseline SBP (mmHg) | 131.6±15.0 | 124.5±14.1 |
| **Procedural** | | |
| Duration of surgical procedure (min) | 52.5 (42.0–67.0) | 62.0 (46.0–74.0) |
| Duration of anesthesia (min) | 81.0 (64.0–92.0) | 89.0 (74.0–102.0) |
| **Analgesic long-term medications secondary to burns** | | |
| Gabapentin | 2 (8.3%)[1] | 5 (20.0%)[2] |
| Tramadol | 10 (41.7%)[1] | 12 (48.0%)[2] |
| Paracetamol-opium | 4 (16.7%)[1] | 4 (16.0%)[2] |
| Short-acting oral morphine | 1 (4.2%)[1] | 2 (8.0%)[2] |
| Long-acting oral morphine | 1 (4.2%)[1] | 2 (8.0%)[2] |
| Amitriptyline | 1 (4.2%)[1] | 1 (4.0%)[2] |
| Ketoprofren | 0 (0.0)[1] | 0 (0.0)[2] |

Values are number (%) for categorical variables and median (interquartile range) or mean±standard error for quantitative variables.

[1]Calculated on 24 patients. [2]Calculated on 25 patients.

Abbreviations: ASA=American Society of Anesthesiologists; HR=heart rate; SBP=systolic blood pressure.

The design of this trial aimed to assess the safety and efficacy of automatic remifentanil administration. In the absence of a gold standard, normalized overall remifentanil administration provided an objective and quantifiable measure for comparison with standard practice. We posited that clinicians would consider automated administration software favorably only if it led to similar or lower administration than standard practice. Hemodynamic stability was another "common sense" choice, as its definition and measurements are easily defined. We were actually surprised by the significant reduction in the administered remifentanil amount in the automatic group, and were under the impression that the decrease is actually relatively low and may be clinically irrelevant. Patients presented with similar pain levels after surgery, and analgesic requirements were also similar in both groups. The trend toward lower VAS from H0+45min in the automatic group was

**Table 2. Primary outcome: Cumulative remifentanil administration during anesthesia.**

| | Standard Practice Group | Automatic Group | Standardized Difference (95% CI) | *% difference* | P-Value |
|---|---|---|---|---|---|
| **Intention-to-treat analysis** | n=26 | n=26 | | | |
| remifentanil (μg.kg$^{-1}$.min$^{-1}$) | 0.152 [0.137–0.237] | 0.125 [0.106–0.149] | −0.88 (−1.46−−0.31) | −17.8 | **0.004** |
| **Per protocol analysis** | n=24 | n=26 | | | |
| remifentanil (μg.kg$^{-1}$.min$^{-1}$) | 0.152 [0.137–0.229] | 0.125 [0.106–0.149] | −0.89 (−1.47−−0.30) | −17.8 | **0.006** |

All measurements were made on Ideal Body Weigth. Standardized differences calculated on rank-transformed data. *P*-values calculated using MW-U test.

not statistically significant, and may have resulted from a lack of statistical power, as it was only a secondary endpoint. Side effects such as nausea and vomiting were infrequent in both groups.

From a methodological point of view, using Minto's pK/pD model for remifentanil administration in the standard practice group vs. an adaptative flow rate (μg.kg$^{-1}$.min$^{-1}$) in the automated group may seem counterintuitive. We acknowledge that this choice was made for practical reasons. First, our unit is well accustomed to Minto's pK/pD model for remifentanil administration and Ce adaptative changes guided by ANI monitoring. The alternative of manually adapting a flow rate infusion appeared much more difficult to implement, and was not a methodological necessity as the chosen endpoints focused only on administered remifentanil between predetermined time points related to surgical stimulation. Second, our unit always proceed during induction of anesthesia as we described in the method section for the standard practice group: remi Ce target set at 6 ng.ml until orotracheal intubation, then lowered to 3 and adapted to maintain the ANIm in the [50–70] target. As the amount of administered remifentanil would depend on the duration of the induction procedure, in particular the ease of tracheal intubation, the resulting amount of remifentanil administered during the induction phase of anesthesia could not precisely be anticipated. On the other hand, the "automatic group" needed an easily implemented remifentanil induction protocol, for which we had experience from the early phase of remifentanil clinical availability: a 1 μg.kg$^{-1}$ bolus was standard procedure before TCI was available. Similarly, 0.1 μg.kg$^{-1}$.min$^{-1}$ infusion dose after induction was considered enough for situations when nociception would be low, typically before the start of any surgical procedure. Considering the whole duration of anesthesia, around 90 min, it was our belief that the difference in method would not bias the results. Extrapolation of the "algorithm for automation" was also of concern: using simple calculations and dosage requirements based on the patient's ideal body weight enables other teams in various surgical settings and various patient populations to reproduce our method. Third, remifentanil administration was normalized and expressed in μg.kg$^{-1}$.min$^{-1}$ in both groups, making comparisons between groups easy without the need of implementing complex statistics about Cp/Ce values over time. Last, the expert-system for automatic remifentanil infusion control does not need to process Minto's pK/pD model for flow rate adaptation: the continuous display of theoretical Cp/Ce concentrations (Minto's model) by the expert system was intended for contextual supervision only.

The applicability of these results to other types of surgery remains to be verified, even if ANI monitoring is unspecific as to surgical site or technique—burn patients often present with amplified but unspecific nociceptive reactions. Any surgical setting is appropriate for future clinical validation.

The proportion of time ANIm spent in the target window was similar in both groups (46% vs. 48%, *p*=0.90), which was surprising because the algorithm targeted the [50–75] range. Likewise, ANIm values over 70 and below 50 were similar in both groups. Liu et al. and Hemmerling et al. have argued that MDPE, MDAPE, and wobble values may be misleading if taken alone when assessing the benefit of closed-loop devices for opioid administration [23,30]. It is our belief that the clinical performance of the ANI-REMI-LOOP system is better understood by clinicians when comparing normalized remifentanil administration and the hemodynamic response of patients to surgery than with measurements specific of automatic closed-loop regulations.

**Table 3. Secondary outcomes during anesthesia (Intention-to-treat analysis).**

| | Standard Practice Group (n = 26) | Automatic Group (n = 26) | Effect-size (95% CI) | P value |
|---|---|---|---|---|
| Number of patient with Hemodynamic reactivity, hypotension, or bradycardia | 8 (30.8) | 12 (46.2) | 1.50 (0.73–3.06)[1] | 0.25 |
| Number of hemodynamic events | 27 | 21 | | |
| Bradycardia | 3 | 0 | | |
| Hypotension | 4 | 2 | | |
| Event duration (minutes) | 2 (1-6) | 1 (1-2) | | **0.038** |
| Duration of Hemodynamic reactivity, hypotension, or bradycardia (minutes) | 10.5 (3.75-25) | 2 (2-3) | | **0.005** |
| Duration relative to intervention duration (%) | 19.4 (6.9–59.9) [8] | 4.2 (2.5–5.7) [12] | −1.43 (−2.43–0.43) | **0.010** |
| Use of ephedrine | 6 (23.1) | 6 (23.1) | 1.00 (0.37–2.70)[1] | 1.00 |
| Total dose of ephedrine (mg) | 9 (6–12) [6] | 9 (6–12) [6] | | |
| Time spent in ANIi intervals (%) | | | | |
| < 50 | 17 (12–28) [25] | 19 (11–36) [25] | 0.25 (−0.31–0.81) | 0.38 |
| 50–70 | 40 (27–45) [25] | 39 (29–43) [25] | −0.08 (−0.64–0.48) | 0.79 |
| > 70 | 40 (27–57) [25] | 39 (21–54) [25] | −0.17 (−0.79–0.39) | 0.55 |
| Time spent in ANIa intervals (%): | | | | |
| < 50 | 15 (7–21) [25] | 15 (8–27) [25] | 0.13 (−0.43–0.69) | 0.65 |
| 50–70 | 46 (25–58) [25] | 46 (32–54) [25] | 0.08 (−0.48–0.64) | 0.79 |
| > 70 | 36 (23–60) [25] | 36 (14–58) [25] | −0.14 (−0.70–0.42) | 0.62 |
| Time spent in BIS intervals (%): | | | | |
| < 40 | 56 (15–80) | 51 (25–69) [25] | 0.02 (−0.54–0.57) | 0.96 |
| 40–60 | 43 (20–82) | 37 (24–71) [25] | −0.14 (−0.69–0.42) | 0.62 |
| > 60 | 1 (0–4) | 2 (0–7) [25] | 0.30 (−0.26–0.85) | 0.29 |
| Cumulative propofol dose during anesthesia (mg.kg-1.min-1) | 0.111 (0.096–0.126) | 0.115 (0.098–0.162) | 0.35 (−0.21–0.90) | 0.22 |
| Propofol target changes (n) | 4 (3–11) | 4 (2–8) | 1.18 (0.71–1.96)[2] | 0.52 |

All measurements were made on Ideal Body Weigh. Values are number (%), median (25th to 75th percentile) or mean ± standard deviation. For quantitative variables, the number of available cases is reported in []. Effect sizes are standardized differences except

[1] relative risk and

[2] risk ratio. CI = confidence interval.

In our RCT, the proportion of time when BIS was on target was quite low in both groups, at 37% in the expert-system automatic group vs. 43% (*p* = 0.62). In the expert-system automatic group, 51% of the time BIS was below 40 vs. 56% in the standard practice group, which indicated an excessive depth of hypnosis overall, even if overall propofol administration was relatively low, and similar to that of other clinical trials [23].

Several meta-analyses addressing ANI monitoring during general anesthesia have expressed a lack of good-quality RCTs [11–13]. Although several clinical trials have validated the safety of ANI opioid guidance in standard practice

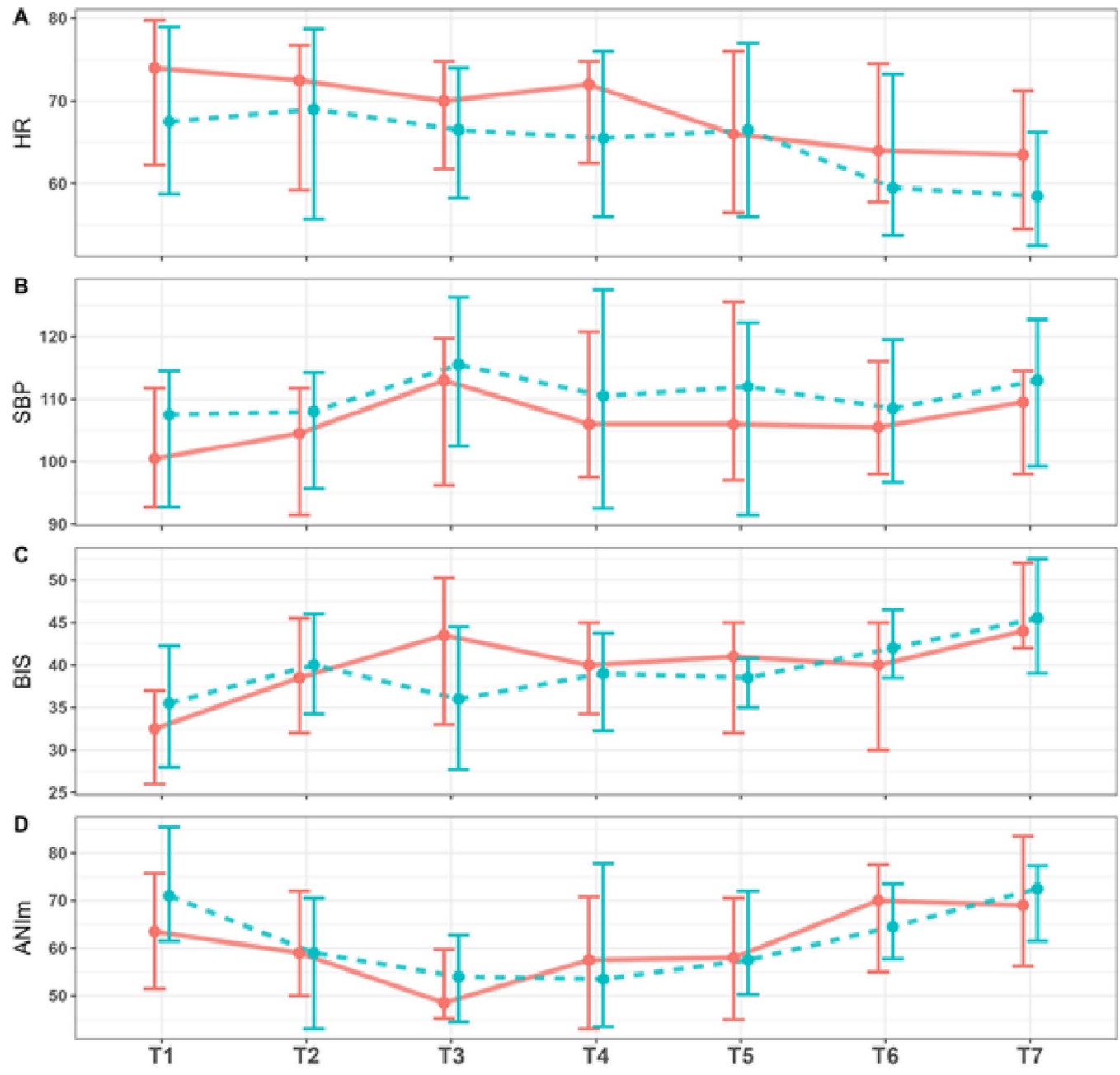

**Fig 3. Overview of hemodynamic, BIS, and ANIm during anesthesia (T1 to T7).** Red lines: Automatic administration, blue lines: standard practice. **Measuring points:** (T1) 5 minutes before start of surgical procedure; (T2) 1 minute, (T3) 5 minutes, (T4) 20 minutes after initiating the surgical procedure; (T5) end of surgery, (T6) during wound dressing, and (T7) patient unstimulated before the awakening phase. Data presented as median [1st; 3rd quartile]. **Abbreviations:** HR = heart rate (bpm); SBP = systolic blood pressure (mmHg); BIS = BiSpectral Index; ANIm = Analgesia Nociception Index.

settings, findings in terms of reduced intra procedural opioid consumption, post-operative pain, or post-operative analgesic sparing effect are often contradictory [14–16,19]. Similar results apply to other so-called "nociception monitors" [1]. To the best of our knowledge, none of these monitors has been used in an algorithm for automatic opioid infusion during general anesthesia. Several dual-loop systems for propofol and remifentanil administration based on EEG signals have been

**Table 4. Secondary outcomes two hours after anesthesia (intention-to-treat analysis).**

| | Standard Practice Group (n = 26) | Automatic Group (n = 26) | Effect size (95% CI) | P value |
|---|---|---|---|---|
| VAS, H0 + 15 min | 60 (40–60) [25] | 60 (50–70) | 0.20 (−0.36–0.75) | 0.48 |
| VAS, H0 + 30 min | 50 (40–70) [25] | 57 (33–70) | −0.04 (−0.59–0.52) | 0.90 |
| VAS, H0 + 45 min | 50 (40–60) [25] | 40 (20–50) [25] | −0.42 (−0.99–0.14) | 0.14 |
| VAS, H0 + 1 h | 40 (30–60) | 30 (10–43) [25] | −0.55 (−1.12–0.01) | 0.055 |
| VAS, H0 + 1 h 15 min | 30 (20–40) [25] | 20 (11–40) [25] | −0.32 (−0.88–0.24) | 0.26 |
| VAS, H0 + 1 h 30 min | 20 (15–30) | 16 (10–30) [25] | −0.43 (−0.99–0.13) | 0.13 |
| VAS, H0 + 1 h 45 min | 20 (10–28) | 10 (10–20) [25] | −0.40 (−0.96–0.16) | 0.16 |
| VAS, H0 + 2 h | 16 (10–20) | 10 (0–20) [25] | −0.22 (−0.78–0.34) | 0.44 |
| Median morphine dose administered (mg) (25th to 75th percentiles) | 10 (5–12) | 10 (5–12) | 0.11 (−0.44–0.66) | 0.69 |
| Administration of ketamine (n) | 9 (34.6%) | 10 (38.5%) | 1.11 (0.54–2.28)[1] | 0.77 |
| median ketamine dose administered (mg) (25th to 75th percentiles) | 20 (20–20) [9] | 20 (20–20) [10] | NA | NA |
| Nausea (n) | 0 | 1 (3.9%) | NA | NA |
| Vomiting (n) | 0 | 0 | NA | NA |

Secondary outcomes, intention-to-treat analysis. Values are number (%), median (25th to 75th percentile) or mean ± standard deviation. For quantitative variables, the number of available cases are reported in []. Effect sizes are standardized differences except

[1]Relative risk and

[2]Risk ratio. VAS = Visual Analog Scale for pain; CI = confidence interval; NA = not applicable

described, and their reliability has been demonstrated [31,32], but none has used a physiological signal directly related to the NAN balance. As pointed out in an editorial by Kuck and Johnson, automation of anesthesia should be safe, transparent, and reduce the workload by enabling the anesthetist to focus on more important and less automatable tasks [33]. An expert system should be "transparent" for the supervisor, who needs to understand why it is driving an opioid flow rate up or down, so that adequate safety can be exercised by stopping the automatic mode [34]. It is our belief that the ANI-REMI-LOOP software is "transparent" and can easily be understood by any anesthetist. Our findings can probably be explained by the capability of the electronic controller to dynamically adapt ("fine tune") opioid delivery to ANI monitoring. The mere number of remi flow rate changes during surgery in the automatic group compared with the number of target changes in the standard practice group gives an idea of the superiority of automation over manual target setting (57 [24–100] in the expert-system automatic group vs 7 [4.5–14]). The superiority of automation over manual administration has been extensively discussed by Coekelenbergh et al. The authors describe automatic and closed-loop devices for the various sensors routinely used during surgical procedures under anesthesia provided by hypnotic drugs, myorelaxants, opioids as well as fluids and vasopressors. The authors stress that automation is safe, reduces the workload of simple repetitive tasks while enhancing compliance with protocols and help improve care [34].

There are limitations to our study. For example, the selection of study participants spanned four years due to a low number of cases and to the COVID pandemic which stopped inclusions. The described trial was only single blind, but there was no way for clinicians to interfere with the ANI-REMI-LOOP device. Applicability to other surgical settings is not

necessarily straightforward, but the type of surgery is probably irrelevant for the validation of this software, as nociception is irrespective of surgical site and technique.

## Conclusion

The clinical efficacy and safety of an expert system based on ANI monitoring (MDoloris technology) for the automatic administration of remifentanil during propofol anesthesia has been demonstrated in a monocentric RCT, leading to significantly reduced remifentanil administration and less hemodynamic events. Pain and analgesic requirements in PACU at 2-h follow-up were not different between the groups. Automatic delivery of an optimal dose of remifentanil at precisely the right time using the ANI-REMI-LOOP device could allow each nociceptive stimulus to be controlled with a minimum dose of opioid while reducing the incidence and duration of hemodynamic events.

## Supporting information

**S1 File. Supplement digital content Table 5. Secondary outcomes during anesthesia (per protocol analysis).**
(DOCX)

**S2 File. Supplement digital content Table 6. Secondary outcomes two hours after anesthesia (per protocol analysis).**
(DOCX)

**S3 File. CONSORT-2010-Checklist.**
(DOC)

**S4 File. PROTOCOLE ani remi loop-FR.**
(DOCX)

**S5 File. PROTOCOLE ani remi loop-EN-revised.**
(DOCX)

**S6 File. DataFigure3.**
(XLSX)

**S7 File. DataBaseANIremiLOOP.**
(XLSX)

## Acknowledgments

The authors would like to thank all care providers at the Burn Center for their kind help with this clinical trial.

## Author contributions

**Conceptualization:** Julien De jonckheere, Mathieu Jeanne.

**Funding acquisition:** Julien De jonckheere.

**Investigation:** Maxence Hureau, Mathilde Herbet, Mathieu Jeanne.

**Methodology:** Julien De jonckheere, Emeline Caillau, Julien Labreuche, Mathieu Jeanne.

**Software:** Julien De jonckheere, Mathieu Jeanne.

**Supervision:** Julien De jonckheere, Maxence Hureau, Mathieu Jeanne.

**Validation:** Julien De jonckheere, Maxence Hureau, Emeline Caillau, Julien Labreuche, Mathieu Jeanne.

**Writing – original draft:** Julien De jonckheere, Maxence Hureau, Emeline Caillau, Benoit Tavernier, Mathieu Jeanne.

**Writing – review & editing:** Julien De jonckheere, Maxence Hureau, Benoit Tavernier, Mathieu Jeanne.

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
