## [Decision Letter · Decision Letter 0]

17 Dec 2024

PONE-D-24-44664Clinical efficacy and safety of automatic remifentanil administration based on Analgesia Nociception Index monitoring during general anesthesia: A randomized controlled clinical trialPLOS ONE

Dear Dr. De jonckheere,

Thank you for submitting your manuscript to PLOS ONE. After careful consideration, we feel that it has merit but does not fully meet PLOS ONE’s publication criteria as it currently stands. Therefore, we invite you to submit a revised version of the manuscript that addresses the points raised during the review process.

Please submit your revised manuscript by Jan 31 2025 11:59PM. If you will need more time than this to complete your revisions, please reply to this message or contact the journal office at plosone@plos.org . Please include the following items when submitting your revised manuscript:

We look forward to receiving your revised manuscript.

Kind regards,

Stefano Turi

Academic Editor

PLOS ONE

Journal Requirements:

“This research was funded by a grant of € 17500 from the APICIL Foundation (Lyon, France).”

3. Please note that funding information should not appear in the Acknowledgments section or other areas of your manuscript. We will only publish funding information present in the Funding Statement section of the online submission form. Please remove any funding-related text from the manuscript. 

“J.D.J. and M.J. are scientific advisers for and own shares of MDoloris Medical Systems, Loos, France. M.Hu., E.C., J.L., M.He,. and B.T. declare no competing interests related to this research.”

6. We note that the original protocol file you uploaded contains a confidentiality notice indicating that the protocol may not be shared publicly or be published. Please note, however, that the PLOS Editorial Policy requires that the original protocol be published alongside your manuscript in the event of acceptance. Please note that should your paper be accepted, all content including the protocol will be published under the Creative Commons Attribution (CC BY) 4.0 license, which means that it will be freely available online, and any third party is permitted to access, download, copy, distribute, and use these materials in any way, even commercially, with proper attribution.

Therefore, we ask that you please seek permission from the study sponsor or body imposing the restriction on sharing this document to publish this protocol under CC BY 4.0 if your work is accepted. We kindly ask that you upload a formal statement signed by an institutional representative clarifying whether you will be able to comply with this policy. Additionally, please upload a clean copy of the protocol with the confidentiality notice (and any copyrighted institutional logos or signatures) removed.

7. We note that the original protocol that you have uploaded as a Supporting Information file contains an institutional logo. As this logo is likely copyrighted, we ask that you please remove it from this file and upload an updated version upon resubmission.

Reviewers' comments:

Reviewer's Responses to Questions

**Comments to the Author**

1. Is the manuscript technically sound, and do the data support the conclusions?

Reviewer #1: Yes

Reviewer #2: Yes

Reviewer #3: Yes

2. Has the statistical analysis been performed appropriately and rigorously? 

Reviewer #1: Yes

Reviewer #2: Yes

Reviewer #3: Yes

3. Have the authors made all data underlying the findings in their manuscript fully available?

Reviewer #1: No

Reviewer #2: Yes

Reviewer #3: Yes

4. Is the manuscript presented in an intelligible fashion and written in standard English?

Reviewer #1: Yes

Reviewer #2: Yes

Reviewer #3: Yes

5. Review Comments to the Author

Reviewer #1: Thank you for reporting your findings.

Abstract: in the results, what quantity is the total decision for the estimates? State it is 95% confidence intervals or interauartile range the first time it is shown in the abstract if this is the case.

Reviewer #2: Dear author(s)

First I congratulate you for your successful, well designed randomized controlled study entitled ‘Clinical efficacy and safety of automatic remifentanil administration based on Analgesia Nociception Index monitoring during general anesthesia: A randomized controlled clinical trial’. However I have some comments.

1)The title accurately reflects the focus of the study, describing the intervention (automatic remifentanil administration), the context (based on the Analgesia Nociception Index), and the study design (a randomized controlled clinical trial) but could include the patient population or surgical context (e.g., burn surgery) to further enhance specificity.

2) The abstract highlights the study's main findings, including reduced remifentanil dosage and maintained safety. It effectively communicates the significance of findings, especially the applicability of ANI monitoring. But you can provide additional context on the potential broader applications of the technology to strengthen the abstract's impact.

3)The introduction outlines the need for improved nociception monitoring and provides background on ANI. It justifies the study by referencing the limitations of current methods and describes the novel ANI-REMI-LOOP system. I think , this section is well-written, with clear context on the problem and study rationale.The hypothesis and objectives are well-stated, aligning with the study design. You can add a brief mention of how this research contributes to existing literature or fills identified gaps.

4)The objectives are stated indirectly within the introduction, focusing on assessing the efficacy and safety of ANI-based remifentanil administration.The study’s objectives are clear but could be outlined in a dedicated section for ease of reference. Please include a standalone section for explicitly stated objectives to improve structure and accessibility.

5)The methodology is robust, describing randomization, ethical considerations, intervention specifics, and outcome measures. Statistical analyses are detailed and appropriate for the study design. However You have to provide additional details on how external validity was ensured, particularly for generalizing findings to other surgical types.

6)The results demonstrate a significant reduction in remifentanil use and improved hemodynamic stability in the automatic group, without compromising safety or postoperative outcomes. Data is presented clearly with adequate use of tables and figures. In this section also you can emphasize clinical relevance of findings to make the benefits more relatable for practitioners.

7)The discussion contextualizes the findings within existing literature, emphasizing the benefits and potential limitations of ANI-based automatic remifentanil administration.The discussion of ANI’s potential in broader surgical settings is particularly strong. But please elaborate on alternative explanations for findings and explore how automation might reduce anesthetist workload.

8)The conclusions summarize the study's key findings, highlighting safety, efficacy, and potential for broader application. Please reiterate the significance of findings for clinical practice and patient outcomes.

9)For Table 1, please consider reformatting the table to separate demographic and clinical characteristics from procedural data for better readability. Also you can add a column for p-values where applicable, comparing the control and automatic groups for baseline variables.

10) For Table 2, please add a column explicitly showing the percentage reduction in remifentanil usage between groups for easier clinical interpretation and include a note clarifying whether the ITT and PP results differ due to excluded patients (e.g., patients with major protocol deviations).

11) For Table 3, draw attention to the most clinically significant differences (e.g., time spent in hemodynamic reactivity) using bold font or separate headings. A brief note summarizing the main findings of this table could help contextualize the results. For example, “The automatic group exhibited significantly reduced time in hemodynamic reactivity, but other outcomes showed no significant differences.”

12) Please emphasize in the manuscript text the potential clinical benefits of trends toward lower pain scores in the automatic group, even if not statistically significant.Discuss whether the pain score differences at later time points (e.g., H0 + 1h, H0 + 1h 45min) are likely clinically meaningful. Also you can include a brief note on how adverse events were assessed and whether the single case of nausea in the automatic group may be attributable to the intervention.

Thank you.

Reviewer #3: The authors present the results of a randomized study conducted in patients scheduled for excision of burns and skin grafts. The aim of the study was to assess the safety and the efficacy of an innovative closed-loop system to automatically deliver remifentanil based on the continuous monitoring of hemodynamic parameters and of the ANI. The study is very original, and the results correspond to a proof-of-concept of the innovation tested. The manuscript is well written and easy to read. Figures and tables are of high quality and adequate. Nonetheless, the manuscript raised the following concern.

MAJOR CONCERNS

1. Methods: page 6: the closed-loop system defines as the 1st priority to protect the patient’s safety. The priority cannot be bypassed. That is a good thing for the patients, but it raises several concerns. This priority appears to be very close to the “old” management of anesthesia that was based on the fluctuations of hemodynamic parameters to adapt the infusion rate of anesthetic drugs. Considering that point, what is the added value of using ANI to monitor intraoperative analgesia? By analogy, BIS monitoring allows to differentiate hemodynamic impairment related to anesthesia overdosage from other causes of hemodynamic impairment. I guess ANI would be useful to apply the same reasoning in the management of intraoperative analgesics. Finally, is the closed-loop system designed to provide adequate intraoperative analgesia or to optimize intraoperative hemodynamics? Please comment that point and discuss the potential impact on results.

2. Eligible patients were burned patients. How many of these patients were treated with beta-blockers to reduce the hypermetabolic response? Could the treatment with beta-blockers impact the measurement of the ANI?

3. Page 7: exclusion criteria: how did you define “sustained tachycardia” and “impaired blood pressure”?

4. Methods: page 8-9: the protocol of induction of anesthesia differs between the 2 groups. Could the authors explain how they proceed to define this protocol for induction and comment on the potential impact of this difference in the management of patients on the results of the study? In particular, the remifentanil dose for induction appears to be higher in the manual group.

5. Methods: one of the main concerns is the protocol of anesthesia in the manual group. How did the authors proceed to define this protocol of management? What was the experience of the investigators in this anesthetic protocol (learning curve)? The sample size is quite small with 25 patients in each group, and it could have been difficult to adequately apply a new anesthetic protocol in patients included in the study. Did the authors measure the good applicability of the anesthetic protocol and the rate of deviance to the anesthetic protocol? Please comment that points and their potential impact on the results of the study.

6. Results: page 12: what is the clinical relevance of the intergroup difference observed on the intraoperative remifentanil dose? In particular, the results suggest that it had no impact on early postoperative analgesia. Is the conclusion of the study: “no matter with how intraoperative analgesia is managed, the main concern is hemodynamics”?

7. Results: what is the potential impact on prognosis to decrease the cumulative time of hemodynamic reactivity, hypotension or bradycardia? The results of the outcome are presented as percentage of time spent, but it could be informative to have these results as absolute time spent with hemodynamic reactivity, hypotension or bradycardia in min, and the duration of the longer episode of hemodynamic reactivity, hypotension or bradycardia in min in each group. This supplemental data would help to assess the clinical relevance of this result.

8. Figure 3: the difference observed on hemodynamics seem to be mainly related to bradycardia. Is this correct? If yes, one explanation could be that investigators in the manual group could have been more concern by hypotension than tachycardia, especially when the decrease in heart rate did not reach critical thresholds. Please comment that point.

9. In the automatic group, the investigator was assisted by an engineer to manage intraoperative administration of remifentanil. Did the authors think that the presence of an additional member of the anesthesia team (the engineer) in the operating room could have impacted the quality of intraoperative administration of remifentanil? Please comment.

10. Table 3: please reformulate the 2nd line: “time spent relative to intervention duration”. What “time spent” refers to?

6. PLOS authors have the option to publish the peer review history of their article (what does this mean? ). If published, this will include your full peer review and any attached files.

**Do you want your identity to be public for this peer review?** For information about this choice, including consent withdrawal, please see our Privacy Policy .

Reviewer #1: No

Reviewer #2: **Yes: ** Fazli Yanik

Reviewer #3: No

---

## [Author Response · Author response to Decision Letter 0]

29 Jan 2025

Dear reviewers,

We thank you for your help in improving the quality of our manuscript. You’ll find our detailed answers below, with the corresponding changes in the manuscript. Modified parts of the manuscript have been highlighted for transparency.

In view of your questions, two additional bibliographic references have been added.

Sincerely

Reviewer #1: Thank you for reporting your findings.

Abstract: in the results, what quantity is the total decision for the estimates? State it is 95% confidence intervals or interquartile range the first time it is shown in the abstract if this is the case.

Thank you for pointing this lack of precision. Results are expressed in median[1st ; 3rd quartile). This has been notified in the abstract, at the end of the method. “A p value < 0.05 was considered statistically significant. Data are presented as median [1st to 3rd quartile].”

Reviewer #2:

Dear author(s)

First I congratulate you for your successful, well designed randomized controlled study entitled ‘Clinical efficacy and safety of automatic remifentanil administration based on Analgesia Nociception Index monitoring during general anesthesia: A randomized controlled clinical trial’.

We thank you for your kind assessment of our work.

However I have some comments.

1) The title accurately reflects the focus of the study, describing the intervention (automatic remifentanil administration), the context (based on the Analgesia Nociception Index), and the study design (a randomized controlled clinical trial) but could include the patient population or surgical context (e.g., burn surgery) to further enhance specificity.

Thank you for pointing the lack of precision. The title has been changed for: “Clinical efficacy and safety of automatic remifentanil administration based on Analgesia Nociception Index monitoring during burn surgery under propofol anesthesia: A randomized controlled clinical trial”

2) The abstract highlights the study's main findings, including reduced remifentanil dosage and maintained safety. It effectively communicates the significance of findings, especially the applicability of ANI monitoring. But you can provide additional context on the potential broader applications of the technology to strengthen the abstract's impact.

Thank you. The conclusion has been amended as you suggested: “Automatic remifentanil administration demonstrated good clinical performances during propofol anesthesia for burn surgery. It is likely that these results can be extrapolated to any surgical setting under general anesthesia, but this needs to be tested with further randomized clinical trials.

3) The introduction outlines the need for improved nociception monitoring and provides background on ANI. It justifies the study by referencing the limitations of current methods and describes the novel ANI-REMI-LOOP system. I think, this section is well-written, with clear context on the problem and study rationale.The hypothesis and objectives are well-stated, aligning with the study design. You can add a brief mention of how this research contributes to existing literature or fills identified gaps.

Thank you. The introduction has been enriched: “Several authors have described automatic opioid delivery during general anesthesia, but the lack of a clear physiological nociceptive signal has led to indirect assessments of nociception, often based on pharmacodynamic interaction models, hemodynamic reactions, or electroencephalographic signals related to cortical arousal [19–21]. In most studies, automated delivery of propofol and remifentanil was shown to be effective and safe, even if the delivery control strategies were rather simplistic, based on proportional, integral, derivative (PID) controllers, or controlling two pharmacological compounds (typically hypnotics and opioids) with a single input variable derived from a simplified EEG signal (e.g. BiSpectral® index) which is neither sensitive nor specific of nociception. In order to improve the performance of remifentanil automatic administration, we have recently designed an expert system for the automatic administration of remifentanil during general anesthesia, whose decision rules for modifying remifentanil flow rate are based on two objectives: i) maintaining stable hemodynamics by avoiding hypertension or tachycardia related to nociception as well as avoiding hypotension and bradycardia and ii) maintaining a relative predominance of paraS activity as assessed by ANI by dynamically increasing or decreasing the remifentanil infusion rate [22, 23].”

4) The objectives are stated indirectly within the introduction, focusing on assessing the efficacy and safety of ANI-based remifentanil administration. The study’s objectives are clear but could be outlined in a dedicated section for ease of reference. Please include a standalone section for explicitly stated objectives to improve structure and accessibility.

Thank you. A standalone section at the end of the introduction section has been added: “The objectives of the study were to evaluate the potential benefit of automatic remifentanil administration on remifentanil or propofol consumption, hemodynamic stability, ANI stability and postoperative pain. “

5) The methodology is robust, describing randomization, ethical considerations, intervention specifics, and outcome measures. Statistical analyses are detailed and appropriate for the study design. However, you have to provide additional details on how external validity was ensured, particularly for generalizing findings to other surgical types.

Thank you. Additional details have been provided at the beginning of the Material & methods chapter with three methodological references conducted with three different nociception monitors. “Several randomized clinical trials have been carried out in various surgical settings to evaluate the benefit of guiding the administration of remifentanil with nociception/antinociception balance monitors [16-18]. The designs and objectives of these studies are relatively standardized, we therefore designed the methodology of our clinical trial on the existing literature.”

6) The results demonstrate a significant reduction in remifentanil use and improved hemodynamic stability in the automatic group, without compromising safety or postoperative outcomes. Data is presented clearly with adequate use of tables and figures. In this section also you can emphasize clinical relevance of findings to make the benefits more relatable for practitioners.

Thank you. The clinical relevance of the results has been further underlined. “The median total remifentanil amount was significantly lower in the expert-system automatic group at 0.125 µg.kg-1.min-1 vs. 0.152 µg.kg-1.min-1 in the standard practice group, with a large effect size (standardized difference = –0.88), corresponding to 17.8% dose reduction (Table 2). The significant reduction in opioid delivery most probably resulted from the specific algorithm fine-tuning remifentanil delivery to the dynamic changes of ANI. Furthermore, the automatic group exhibited significantly reduced time spent with hemodynamic reactivity while other secondary outcomes showed no significant differences between groups (Table 3).”

7) The discussion contextualizes the findings within existing literature, emphasizing the benefits and potential limitations of ANI-based automatic remifentanil administration.The discussion of ANI’s potential in broader surgical settings is particularly strong. But please elaborate on alternative explanations for findings and explore how automation might reduce anesthetist workload.

Thank you. The discussion section has been enriched as follow: “Our findings can probably be explained by the capability of the electronic controller to dynamically adapt ("fine tune") opioid delivery to ANI monitoring. The mere number of remi flow rate changes during surgery in the automatic group compared with the number of target changes in the standard practice group gives an idea of the superiority of automation over manual target setting (57 [24–100] in the expert-system automatic group vs 7 [4.5–14]). The superiority of automation over manual administration has been extensively discussed by Coekelenbergh et al. The authors describe automatic and closed-loop devices for the various sensors routinely used during surgical procedures under anesthesia provided by hypnotic drugs, myorelaxants, opioids as well as fluids and vasopressors. The authors stress that automation is safe, reduces the workload of simple repetitive tasks while enhancing compliance with protocols and help improve care [32].”

8) The conclusions summarize the study's key findings, highlighting safety, efficacy, and potential for broader application. Please reiterate the significance of findings for clinical practice and patient outcomes.

Thank you. The conclusion has been changed accordingly: “The clinical efficacy and safety of an expert system based on ANI monitoring (MDoloris technology) for the automatic administration of remifentanil during propofol anesthesia has been demonstrated in a monocentric RCT, leading to significantly reduced remifentanil administration and less hemodynamic events. Pain and analgesic requirements in PACU at 2-h follow-up were not different between the groups. Automatic delivery of an optimal dose of remifentanil at precisely the right time using the ANI-REMI-LOOP device could allow each nociceptive stimulus to be controlled with a minimum dose of opioid while reducing the incidence and duration of hemodynamic events.”

9) For Table 1, please consider reformatting the table to separate demographic and clinical characteristics from procedural data for better readability. Also you can add a column for p-values where applicable, comparing the control and automatic groups for baseline variables.

Thank you. Table 1 has been edited accordingly, clearly separating “demographic data”, “clinical data”, “procedural data” and “Analgesic long-term medications secondary to burns”

Regarding the suggestion to provide p-value for between group differences in baseline characteristics, we respectfully disagree in light of the CONSORT recommendation: baseline characteristics should not be compared between groups in randomized controlled trial (RCT) but interpreted regarding the descriptive analysis “Item 15 from CONSORT : Such hypothesis testing is superfluous and can mislead investigators and their readers. Rather, comparisons at baseline should be based on consideration of the prognostic strength of the variables measured and the size of any chance imbalances that have occurred”. As commented by Altman DG, performing a significance test to compare baseline variables in RCT is to assess the probability of something having occurred by chance when we know that it did occur by chance (Altman DG. Comparability of randomized groups. The Statisticien (1985), pp 125-136).

10) For Table 2, please add a column explicitly showing the percentage reduction in remifentanil usage between groups for easier clinical interpretation and include a note clarifying whether the ITT and PP results differ due to excluded patients (e.g., patients with major protocol deviations).

Thank you. An additional column has been added, showing that the percentage reduction is the same following ITT and PP results. It is our belief that excluded patients did not impact the presented results.

11) For Table 3, draw attention to the most clinically significant differences (e.g., time spent in hemodynamic reactivity) using bold font or separate headings. A brief note summarizing the main findings of this table could help contextualize the results. For example, “The automatic group exhibited significantly reduced time in hemodynamic reactivity, but other outcomes showed no significant differences.”

Thank you. Two additional lines have been added to Table 3, indicating the number of hypotensive and bradycardic events. The following text has been added for clarity: “Twenty-one “Hemodynamic reactivity, hypotension, or bradycardia” events occurred in 12 cases (46.2%) in the expert-system automatic group and 27 “Hemodynamic reactivity, hypotension, or bradycardia” events occurred in 8 (30.8%) cases in the standard practice group (p = 0.25). The median duration for one event was 2 [2-3] min for the expert-system automatic group and 2 [1-6] min for the standard practice group (p=0.038), and their cumulative duration was 2 [2-3] versus 10.5 [3.75-25] min respectively (p=0.005). Two hypotension and no bradycardic events were observed in the expert-system automatic group whereas 4 hypotension and 3 bradycardic events occurred in the standard practice group (Table 3).”

12) Please emphasize in the manuscript text the potential clinical benefits of trends toward lower pain scores in the automatic group, even if not statistically significant. Discuss whether the pain score differences at later time points (e.g., H0 + 1h, H0 + 1h 45min) are likely clinically meaningful. Also you can include a brief note on how adverse events were assessed and whether the single case of nausea in the automatic group may be attributable to the intervention.

Thank you. The following sentences have been added in the discussion section: “Patients presented with similar pain levels after surgery, and analgesic requirements were also similar in both groups. The trend toward lower VAS from H0+45min in the automatic group was not statistically significant, and may have resulted from a lack of statistical power, as it was only a secondary endpoint. Side effects such as nausea and vomiting were infrequent in both groups."

Reviewer #3: The authors present the results of a randomized study conducted in patients scheduled for excision of burns and skin grafts. The aim of the study was to assess the safety and the efficacy of an innovative closed-loop system to automatically deliver remifentanil based on the continuous monitoring of hemodynamic parameters and of the ANI. The study is very original, and the results correspond to a proof-of-concept of the innovation tested. The manuscript is well written and easy to read. Figures and tables are of high quality and adequate. Nonetheless, the manuscript raised the following concern.

MAJOR CONCERNS

1. Methods: page 6: the closed-loop system defines as the 1st priority to protect the patient’s safety. The priority cannot be bypassed. That is a good thing for the patients, but it raises several concerns. This priority appears to be very close to the “old” management of anesthesia that was based on the fluctuations of hemodynamic parameters to adapt the infusion rate of anesthetic drugs. Considering that point, what is the added value of using ANI to monitor intraoperative analgesia?

Thank you for raising this question. The safety focus has led us to prioritize hemodynamics (priority 1) over antinociception (priority 2). As presented in the result section, hemodynamic impairment was infrequent in both groups, and represented a small proportion of anesthesia duration when present, so that regulation of remifentanil’s flow was primarily driven by ANI measures (priority 2 rules). We have added two short explanations in the general description of the algorithm: “This rule has been implemented in order to limit the risk of severe hypotension (SBP<70 mmHg): it only allows limiting the infusion flow in case of SBP decrease.” and “Priority 2 rules: dynamic adaptation of remifentanil flowrate. Several studies have demonstrated that variations in ANI enable to anticipate hemodynamic reactivity in relation with the NAN balance. We therefore implemented ANI based decision rules in order to anticipate and possibly avoid hemodynamic reactivity by increasing the infusion flow in order to maintain ANIm slightly over ANImin. Other rules also aimed at decreasing the infusion flow in order to maintain ANI bellow ANImax. .”

2. Eligible patients were burned patients. How many of these patients were treated with beta-blockers to reduce the hypermetabolic response? Could the treatment with beta-blockers impact the measurement of the ANI?

Thank you for raising this point. Our unit mainly uses beta-blockers in patients with Burn Surface Area over 10-15%. No patient included in the presented

---

## [Decision Letter · Decision Letter 1]

21 Mar 2025

Clinical efficacy and safety of automatic remifentanil administration based on Analgesia Nociception Index monitoring during burn surgery under propofol anesthesia: A randomized controlled clinical trial

PONE-D-24-44664R1

Dear Dr. De jonckheere ,

We’re pleased to inform you that your manuscript has been judged scientifically suitable for publication and will be formally accepted for publication once it meets all outstanding technical requirements.

Kind regards,

Stefano Turi

Academic Editor

PLOS ONE

Additional Editor Comments (optional):

Reviewers' comments:

Reviewer's Responses to Questions

**Comments to the Author**

1. If the authors have adequately addressed your comments raised in a previous round of review and you feel that this manuscript is now acceptable for publication, you may indicate that here to bypass the “Comments to the Author” section, enter your conflict of interest statement in the “Confidential to Editor” section, and submit your "Accept" recommendation.

Reviewer #1: All comments have been addressed

Reviewer #2: All comments have been addressed

2. Is the manuscript technically sound, and do the data support the conclusions?

Reviewer #1: (No Response)

Reviewer #2: Yes

3. Has the statistical analysis been performed appropriately and rigorously? 

Reviewer #1: (No Response)

Reviewer #2: Yes

4. Have the authors made all data underlying the findings in their manuscript fully available?

Reviewer #1: (No Response)

Reviewer #2: Yes

5. Is the manuscript presented in an intelligible fashion and written in standard English?

Reviewer #1: (No Response)

Reviewer #2: Yes

6. Review Comments to the Author

Reviewer #1: (No Response)

Reviewer #2: Dear authors; I read and reviewed the revised article according to my suggestions.

Thank you for your revised article.

My opinion for revised article is 'Accept'.

7. PLOS authors have the option to publish the peer review history of their article (what does this mean? ). If published, this will include your full peer review and any attached files.

**Do you want your identity to be public for this peer review?** For information about this choice, including consent withdrawal, please see our Privacy Policy .

Reviewer #1: No

Reviewer #2: **Yes: ** Fazli Yanik

---

## [Editor Report · Acceptance letter]

PONE-D-24-44664R1

PLOS ONE

Dear Dr. De jonckheere,

I'm pleased to inform you that your manuscript has been deemed suitable for publication in PLOS ONE. Congratulations! Your manuscript is now being handed over to our production team.

Kind regards,

on behalf of

Dr. Stefano Turi

Academic Editor

PLOS ONE